# Cannabinoids, Phenolics, Terpenes and Alkaloids of *Cannabis*

**DOI:** 10.3390/molecules26092774

**Published:** 2021-05-08

**Authors:** Mohamed M. Radwan, Suman Chandra, Shahbaz Gul, Mahmoud A. ElSohly

**Affiliations:** 1National Center for Natural Products Research, School of Pharmacy, University of Mississippi, Oxford, MS 38677, USA; mradwan@olemiss.edu (M.M.R.); suman@olemiss.edu (S.C.); 2ElSohly Laboratories, Inc., 5 Industrial Park Drive, Oxford, MS 38655, USA; swgul@go.olemiss.edu; 3Sally McDonnell Barksdale Honors College, University of Mississippi, Oxford, MS 38677, USA; 4Department of Pharmaceutics and Drug Delivery, School of Pharmacy, University of Mississippi, Oxford, MS 38677, USA

**Keywords:** *Cannabis sativa*, cannabinoids, non-cannabinoid phenols, flavonoids, terpenes, alkaloids

## Abstract

*Cannabis sativa* is one of the oldest medicinal plants in the world. It was introduced into western medicine during the early 19th century. It contains a complex mixture of secondary metabolites, including cannabinoids and non-cannabinoid-type constituents. More than 500 compounds have been reported from *C. sativa*, of which 125 cannabinoids have been isolated and/or identified as cannabinoids. Cannabinoids are C_21_ terpeno-phenolic compounds specific to *Cannabis.* The non-cannabinoid constituents include: non-cannabinoid phenols, flavonoids, terpenes, alkaloids and others. This review discusses the chemistry of the cannabinoids and major non-cannabinoid constituents (terpenes, non-cannabinoid phenolics, and alkaloids) with special emphasis on their chemical structures, methods of isolation, and identification.

## 1. Introduction

*Cannabis sativa* L. belongs to the plant family Cannabaceae, which only has one genus (Cannabis) with only one highly variable species, *C. sativa*. This is one of the oldest plants grown for food, fiber, and medicine. It grows in all habitats, ranging from sea level to temperate to alpine foothills. The plant originated in Western Asia and introduced to western medicine during the early 19th century. Cannabis has a long history of being used as a medicine to treat a variety of ailments, including asthma, epilepsy, fatigue, glaucoma, insomnia, nausea, pain, and rheumatism [1].

Cannabis is primarily a dioecious plant (male and female flowers occur on individual plants); it is only occasionally found as a hermaphrodite (male and female flowers on the same plant). It flowers under a short photoperiod (below 12 h of light) and continues growing vegetatively during the longer photoperiod days.

The plant is a chemically complex species, due to its numerous natural constituents [2]. Cannabinoids, a specific chemical class found in cannabis, are produced in the glandular trichomes of the plant. Among the cannabinoid constituents of cannabis, Δ^9^-tetrahydrocannabinol (Δ^9^-THC), which is naturally present in the form of an acid (Δ^9^-tetrahydrocannabinolic acid, Δ^9^-THCA), is the main psychoactive constituent of the plant. Decarboxylation of the acid with age or heat is required to form the pharmacologically active Δ^9^-tetrahydrocannabinol [3]. Cannabidiol (CBD), another cannabinoid of current interest, is reported to be active as an antiepileptic agent, particularly for the treatment of intractable pediatric epilepsy [4,5].

Other than Δ^9^-THC and CBD, tetrahydrocannabivarin (THCV), cannabinol (CBN), cannabigerol (CBG), and cannabichromene (CBC) are four other major cannabinoids also identified in *C. sativa*. Modern studies report that the pharmacological effects of phytocannabinoids result from their ability to interact with cannabinoid receptors and/or with other kinds of pharmacological targets, including non-cannabinoid receptors [6]. Thus far, more than 500 constituents have been reported from Cannabis, out of which 125 are classified as cannabinoids. The non-cannabinoid constituents include non–cannabinoid phenols, flavonoids, terpenes, alkaloids and others.

The present review discusses the chemistry of all identified major Cannabis constituents including cannabinoid and non-cannabinoid constituents (non–cannabinoid phenols, flavonoids, terpenes, and alkaloids), with special emphasis on the chemical structures, methods of isolation, and identification. This review also updated the chemistry of 125 cannabinoids, 42 phenolics, 34 flavonoids, 120 terpenes and 2 alkaloids. The last review on cannabis chemistry was published by our group in 2017 as a chapter book which focused on the new constituents reported between 2005 and 2017 only but, in this review, we provided a chemical account on the chemistry of major constituents (323 compounds) of cannabis from 1940 up to now along with their structures. For the reader, this review should provide almost all the information of cannabinoids and non-cannabinoids, their methods of isolation, identification and structures in one place. In addition, these 323 compounds include 12 new compounds (five cannabinoids, one cannabispiran, three flavonoids, and three terpenes). The inclusion of all the 323 chemical structures along with all other details makes this review unique.

## 2. Cannabinoids: (125 Compounds)

Cannabinoids are a group of compounds with a characteristic C21 terpenophenolic backbone. This nomenclature can be applied to parent cannabinoids, cannabinoid derivatives, and transformation products. These cannabinoids can be further classified into 11 cannabinoid sub-classes, namely; cannabichromene (CBC), cannabidiol (CBD), cannabielsoin (CBE), cannabigerol (CBG), cannabicyclol (CBL), cannabinol (CBN), cannabinodiol (CBND), cannabitriol (CBT), (−)-Δ^8^-*trans*-tetrahydrocannabinol (Δ8-THC), (−)-Δ^9^-*trans*-tetrahydrocannabinol (Δ^9^-THC), and miscellaneous-type cannabinoids.

### 2.1. (−)-Δ^9^-Trans-Tetrahydrocannabinol (Δ^9^-THC) Type (25 Cannabinoids)

The isolation of pure (−)-Δ^9^-*trans*-tetrahydrocannabinol (Δ^9^-THC, **1**), from a hexane extract of hashish using column chromatography over florisil followed by alumina was reported by Goani and Mecholum in 1964. A crystalline nitrophenylurethane derivative of THC was prepared, followed by a mild alkaline hydrolysis for further purification of THC. IR and NMR spectroscopic techniques were used to elucidate its chemical structure [7]. In 1967, (−)-Δ^9^-*trans*-tetrahydrocannabinolic acid A (Δ^9^-THCAA, **2**) was isolated using a cellulose powder column (eluted with a mixture of hexane and dimethylformamide), followed by preparative thin layer chromatography. The chemical structure of THCAA (**2**) was elucidated through a combination of UV, IR, and NMR spectroscopic analysis [8]. The isolation of Δ^9^-THCAA (**2**) was also reported using an acid–base extraction procedure [9]. The methodology of obtaining both Δ^9^-THC (**1**) and Δ^9^-THCAA (**2**) from *C. sativa* plant material was optimized, using an enhanced extraction procedure, and the separation of Δ^9^-THC (**1**) was further improved using fractional distillation. For extensive separation, different stationary phases, such as silica, alumina, and C_18_ silica, were tested to obtain pure Δ^9^-THC (**1**) with a high yield. Following these tests, an efficient, preparative C_18_ HPLC method was developed for the purification of Δ^9^-THC (**1**) from the distillate. These steps resulted in an overall reduced cost of production [10].

The isolation of (−)-Δ^9^-*trans*-tetrahydrocannabinolic acid B (Δ^9^-THCAB, **3**) was reported from a hashish sole, using a silicic acid column eluted with a mixture of diethyl ether in petroleum ether. The structure of **3** was established by comparing its physical properties (m.p., optical rotation, MS, UV, and IR) with those of THCAA (**2**) [11]. The crystal structure of Δ^9^-THCAB (**3**) was determined in 1975 through slow evaporation of its chloroform solution [12]. The compounds (−)-Δ^9^-*trans*-tetrahydrocannabinol-C_4_ (Δ^9^-THC-C_4_, **4**) and (−)-Δ^9^-*trans*-tetrahydrocannabinolic acid A-C_4_ (Δ^9^-THCAA-C_4_, **5**) were identified and characterized through the analysis of ethyl acetate extracts of police confiscated cannabis resins, tinctures, and leaves using GC-MS [13]. In 1971, (−)-Δ^9^-*trans*-tetrahydrocannabivarin (Δ^9^-THCV, **6**) was isolated from a cannabis tincture of Pakistani origin, using the counter-current distribution technique to isolate the compound from a light petroleum ether extract and its chemical structure was determined by IR, NMR and MS spectroscopy, and was confirmed by synthesis [14]. The isolation of (−)-Δ^9^-*trans*-tetrahydrocannabivarinic acid (Δ^9^-THCVAA, **7**) from fresh *C. sativa* leaves from South Africa was reported in 1973 [15]. The chemical structure of Δ^9^-THCVAA (**7**) was determined in 1977 using IR, UV, and NMR spectral analysis; the structure was confirmed by via MS analysis of the methyl ester of the cannabinoid, yielding a characteristic fragmentation pattern with 28 mass units less [16]. In 1973, (−)-Δ^9^-*trans*-tetrahydrocannabiorcol (Δ^9^-THCO or Δ^9^-THC_1_, **8**) was identified from the analysis of a light petroleum ether extract of Brazilian marijuana [17]. The GC-MS analysis of different cannabis samples resulted in the identification of (−)-Δ^9^-*trans*-tetrahydrocannabiorcolic acid (Δ^9^-THCOAA, **9**) [13].

In 2015, (−)-Δ^9^-*trans*-tetrahydrocannabinal (Δ^9^-THC aldehyde, **10**) was isolated from a high potency variety of *C. sativa* by applying VLC (Vacuum Liquid Chromatography), silica gel column chromatography, and HPLC chromatographic techniques and identified by 1D and 2D NMR [18]. Eight cannabinoid esters (**11**–**17**) were isolated from the hexane extract of the same high potency variety of *C. sativa* seven years earlier, using various chromatographic techniques, such as vacuum liquid chromatography (VLC), C_18_ semi-preparative HPLC, and chiral HPLC. The compounds were identified as β-fenchyl (−)-Δ^9^-*trans*-tetrahydrocannabinolate (**11**), α-fenchyl (−)-Δ^9^-*trans*-tetrahydrocannabinolate (**12**), *epi*-bornyl (−)-Δ^9^-*trans*-tetrahydrocannabinolate (**13**), bornyl (−)-Δ^9^-*trans*-tetrahydrocannabinolate (**14**), α-terpenyl (−)-Δ^9^-*trans*-tetrahydrocannabinolate (**15**), 4-terpenyl (−)-Δ^9^-*trans*-tetrahydrocannabinolate (**16**), α-cadinyl (−)-Δ^9^-*trans*-tetrahydrocannabinolate (**17**), and γ-eudesmyl (−)-Δ^9^-*trans*-tetrahydrocannabinolate (**18)** by spectroscopic analysis (^1^H NMR, ^13^C NMR, 2D NMR) and GC-MS analysis [19]. Three additional cannabinoid-type compounds were isolated and identified in 2015, namely 8α-hydroxy-(−)-Δ^9^-*trans*-tetrahydrocannabinol (**19**), 8*β*-hydroxy-(−)-Δ^9^-*trans*-tetrahydro cannabinol (**20**), and 11-acetoxy- (−)-Δ^9^-*trans*-tetrahydrocannabinolic acid A (**21**) from high potency *C. sativa*. Multiple chromatographic techniques were used, including silica gel VLC, C_18_-solid phase extraction (SPE), and HPLC [20]. The compound 8-oxo-(−)-Δ^9^-*trans-*tetrahydrocannabinol (**22**) was also isolated from the same high potency *C. sativa* variety [18]. The chemical structures of compounds **19**–**20** were elucidated based on HRESIMS and NMR analysis [18,20].

Cannabisol (**23**) was isolated from a group of illicit cannabis samples received under the cannabis potency monitoring program, with a high CBG content using flash silica gel chromatography eluted with hexane/CHCl_3_ (1:1). The structure of **23** was unambiguously deduced by HRESIMS, GCMS, and NMR spectroscopy. GC-MS analysis of these samples indicated the dimeric nature of the compound, displaying two molecular ion peaks at *m/z* 314 and *m/z* 328, corresponding to Δ^9^-THC and 2-methyl-Δ^9^-THC, respectively [21].

Two new heptyl and hexyl homologs of Δ^9^-THC namely, (−)-Δ^9^-*trans*-tetrahydrocannabiphorol (**24**) and (−)-Δ^9^-*trans*-tetrahydrocannabihexol (**25**) were recently isolated from the hexane extract of *C. sativa* inflorescences of an Italian origin (strain CIN-RO). The hexane extract was cooled at −20 °C for 48 h to remove waxes by precipitation. The dewaxed extract was subjected to semi-preparative liquid chromatography on a C18 stationary phase column to isolate compounds **24** and **25** after heating the corresponding acids at 120 °C for 2 h as clear oil. ^1^ H and ^13^C NMR, circular dichroism (CD) and UV absorption spectroscopy, along with LC-HRMS were used to determine their chemical structure. Both compounds were prepared synthetically using stereoselective synthesis to confirm their chemical structures [22,23]. All of the Δ^9^-THC type cannabinoids are shown in Figure 1.

### 2.2. (−)-Δ8-Trans-Tetrahydrocannabinol (Δ^8^-THC) Type (Five Cannabinoids)

The cannabinoid (−)-Δ^8^-*trans*-tetrahydrocannabinol (Δ^8^-THC, **26**) was isolated in 1966 from the leaves and flowers of Cannabis grown in Maryland. Δ^8^-THC was purified from the petroleum ether extract through silicic acid column chromatography using benzene and an eluent [24]. Nine years later, its carboxylic acid, Δ^8^- *trans*-tetrahydrocannabinolic acid A (Δ^8^-THCA, **27**) was isolated as the methyl ester from a Cannabis plant of Czechoslovakian origin (Figure 2) [25]. NMR analysis was used to determine the chemical structure of [26]. In 2015, three hydroxylated Δ^8^-THC-type cannabinoids were isolated from high-potency *C. sativa*. The structural elucidation for 10α-hydroxy-Δ^8^-tetra-hydrocannabinol (**28**), 10β-hydroxy-Δ^8^-tetra-hydrocannabinol (**27**) and 10a-α-hydroxy-10-oxo-Δ^8^-tetrahydrocannabinol (**30**) was carried out using 1D and 2D NMR spectra [18,20]. The Δ^8^-THC type cannabinoids are shown in Figure 2.

### 2.3. Cannabigerol (CBG) Type (16 Cannabinoids)

Sixteen cannabinoids (**31**–**46**) were classified as CBG-type cannabinoids (Figure 3). In 1964, cannabigerol ((*E*)-CBG, **31**) was isolated from cannabis resin, using florisil chromatography; the chemical structure of (*E*)-CBG (**31**) was confirmed through synthesis [7]. Cannabigerolic acid (CBGAA, **32**) and the monomethyl ether of CBGAA (CBGAM, **34**) were isolated in 1975, proving that CBGA (**32**) is the first cannabinoid synthesized in the biosynthetic pathway of Δ^9^-THCAA (**2**) [27]. The monomethyl ether of (*E*)-CBG (CBGM, **33**) was isolated from a benzene extract of hemp by heating the extract with toluene for seven hours and purifying using column chromatography (silica-gel column) with benzene as the eluent [28]. Cannabigerovarin (CBGV, **35**) and cannabigerovarinic acid (CBGVA, **36**) were isolated from a benzene extract of “Meao variant” cannabis from Thailand; the chemical structures for CBGV (**35**) and CBGVA (**36**) were determined through IR, NMR, and UV spectroscopies [16,29]. In 1995, cannabinerolic acid ((*Z*)CBGA, **37**) was isolated from an acetone extract of the leaves of a Mexican strain of *C. sativa*, using silica-gel column chromatography. The chemical structure of (*Z*)CBGA (**37**) was determined through FAB-MS and NMR spectroscopies and confirmed by the synthesis of the cannabinoid [30]. Another cannabinoid derivative, 5-acetyl-4-hydroxy-cannabigerol (**38**), was isolated from the buds of high potency *C. sativa* using normal phase HPLC of the polar fractions [31]. Two esters of CBGA (**32**), γ-eudesmyl-cannabigerolate (**37**) and α-cadinyl-cannabigerolate (**38**), were also isolated from high potency *C. sativa*, using chiral HPLC and their chemical structures were established by GC/MS, HRESIMS, 1D NMR, and 2D NMR [19]. In 2008, four epoxy cannabigerol derivatives, (±)-6,7-*trans*-epoxycannabigerolic acid (**41**), (±)-6,7-*cis*-epoxycannabigerolic acid (**42**), (±)-6,7-*cis*-epoxycannabigerol (**43**), and (±)-6,7-*trans*-epxoycannabigerol (**44**), were isolated from high potency *C. sativa* (grown in Mississippi) by the application of various chromatographic techniques (VLC, flash chromatography, and HPLC). The chemical structures of the four epoxy cannabigerol derivatives (**41**–**44**) based on NMR and HRESIMS spectroscopic analyses [32]. The polar dihydroxycannabigerol derivative (camagerol, **45**) was isolated from the aerial parts of a *C. sativa* strain, Carma, using reverse-phase (C_18_) silica-gel column chromatography, followed by normal-phase silica gel column chromatography and, finally, normal phase (NP)-HPLC. The wax of the aerial parts of the Carma strain was hydrolyzed and purified, using silica and alumina column chromatography, resulting in waxy and non-waxy fractions. The farnesyl prenylogue of cannabigerol (sesquicannabigerol, **46**) was isolated from one of the waxy fractions. Its structure was established on the basis of NMR spectroscopic analysis (^1^H-NMR, ^13^C-NMR) and semisynthesis from cannabigerol (**31**) [33]. The CBG type cannabinoids are shown in Figure 3.

### 2.4. Cannabidiol (CBD) Type (10 Cannabinoids)

Cannabidiol (CBD, **47**) was isolated from an ethanolic extract (red oil) of Minnesota wild hemp. It was purified as a *bis*-3,5-dinitrobenzoate crystalline derivative [34]. The absolute configuration of CBD (**47**), (−)-*trans*-(1*R*,6*R*), was determined in 1969 through synthesis from (+)-*cis*- and (+)-*trans*-*p*-menthadien-(2,8)-ol and olivetol [35]. CBDA-C_5_ (**48**) was isolated from the fresh tops and leaves of *C. sativa* after extraction with benzene and identified by comparing its UV spectrum with that of CBD derivatives [36]. Cannabidiol monomethylether (CBDM-C_5_, **49**) was isolated from the decarboxylated ethanol extract of hemp leaves on using florisil and silica gel column chromatography. CBDM (**49**) was identified by comparing its physical properties with an authentic sample YA [37]. CBD-C_4_ (**50**) was obtained from the ethyl acetate extract of cannabis resin and leaves after derivatization; it was characterized through GC-FID and GC-MS analyses [13]. Cannabidivarin (CBDV-C_3**,**_
**51**) was reported from hashish through silica gel chromatography and was identified by spectroscopic analysis (UV, IR, NMR and EIMS) [38].

Shoyama et al. isolated cannabidivarinic acid (CBDVA, **52**) from the benzene extract of Thai cannabis, which was chromatographed on a polyamide column eluted with H_2_O:MeOH (1:1–1:6). Its structure was elucidated using UV, IR, and ^1^H NMR spectroscopic analysis [16]. Cannabidiorcol (CBD-C_1_, **53**) was identified in the hexane extract of Lebanese hashish by combined gas chromatography–mass spectrometry [39].

Recently, two new CBD homologues with *n*-hexyl and *n*-heptyl side chain were isolated from the hexane extract of *C. sativa* and their chemical structures were assigned as cannabidihexol (CBDH, **54**) and cannabidiphorol (CBDP, **55**). The two compounds were purified using a semi-preparative C18 HPLC using a mixture of ACN/0.1 aqueous formic acid as a mobile phase. The fractions containing **52** and **53** were analyzed by HRESMS. The chemical structures of CBDH and CBDP were determined by HNMR, C-NMR, UV, and HR-ESI-MS and confirmed by stereoselective synthesis [22,23].

In 2020, a CBD dimer was isolated from the hexane extract of hemp and named Cannabitwinol, (CBDD, **56**). The hexane extract was chromatographed on a Sigel column, which was eluted with hexane/CH_2_Cl_2_ followed by semi preparative C18-HPLC using a mixture of ACN/H_2_O/formic acid (7:3:0.1) as the mobile phase. The chemical structure of CBDD (**56**) was determined by applying a plethora of 1D and 2D NMR at −30 °C and was confirmed by HRESIMS and MS/MS spectrometry. The authors confirmed the chemical structure of **56** as two units of CBD connected by a methylene bridge and suggested that the dimerization of CBD occurred as a result of enzymatic reaction medicated by a one-carbon donor enzyme like methylene tetrahydrofolate [40]. All of the CBD type cannabinoids are shown in Figure 4.

### 2.5. Cannabinodiol (CBND) Type (Two Cannabinoids)

Only two cannabinodiol (CBND) type cannabinoids (**57** and **58**) have been reported (Figure 4). In 1972, cannabinodivirin (CBND-C_3_, **57**) was detected in hashish by GC-MS analysis [41]. In 1977, cannabinodiol (CBND-C_5_, **58**) was isolated from Lebanese hasish, using silica-gel column chromatography. The structure of CBND-C_5_ (**58**) was determined by ^1^H-NMR and confirmed by the phytochemical transformation of cannabinol into cannabinodiol [42].

### 2.6. Cannabielsoin (CBE) Type (Five Compounds)

In 1973, cannabielsoin (CBE-C5, **59**) was detected in the ethanolic extract of Lebanese hashish. The extract was subjected to counter current distribution followed by GCMS analysis [43]. In 1974, its configuration was determined to be 5aS, 6S, 9R, 9aR [44]. Both cannabielsoin acid A (CBEAA, **60**) and cannabielsoin acid B (CBEAB, **61**) were isolated from hashish; the structural elucidation was carried by NMR spectroscopy and chemical transformations [45]. Cannabielsoin-C3 (CBE-C3, **62**) and Cannabielsoic acid B-C3 (CBEAB-C3, **63**) were also reported from cannabis [2,46]. The structures of these five cannabinoids (**59**–**63**) can be found in Figure 4.

### 2.7. Cannabicyclol (CBL) Type Three3 Compounds)

Krote and Sieper isolated cannabicyclol (CBL, **64**) from hashish by thin layer chromatography, but its structure was correctly elucidated by Mechoulam and Gaoni in 1967, based on spectral data [47,48]. The relative configuration of CBL (**64**) was determined by X-ray analysis in 1970 [49]. Cannabicyclolic acid (CBLA, **65**) was obtained from the benzene extract of cannabis. The benzene extract was chromatographed on a polyamide column using methanol water as a mobile phase. CBLA was isolated as a methylated derivative and considered to be an artifact formed when CBCA is naturally irradiated during storage [50]. Cannabicyclovarin (CBLV, **66**) was detected in the ether extract of Congo marihuana and was identified by GLC and GCMS [51]. CBLA (**65**) was determined to be produced during the natural irradiation of cannabichromenic acid (CBCA, **68**), proving that CBLA is not a natural cannabinoid [50]. The chemical structures of CBL-type cannabinoids are shown in Figure 4.

### 2.8. Cannabichromene (CBC) Type (Nine Compounds)

Cannabichromene (CBC, **67**) was isolated from the hexane extract of hashish, using column chromatography (florisil column) in 1966 [52]. The carboxylic acid derivative of CBC (**67**), cannabichromenic acid (CBCA, **68**) was isolated from the benzene extract of hemp, using silica-gel column chromatography. The chemical structure of CBCA (**68**) was determined using IR, NMR, and UV spectroscopies [27]. Cannabivarichromene (± CBCV, **69**) was identified through GC-MS analysis, and both cannabichromevarin (+ CBCV, **70**) and cannabichromevarinic acid (CBCVA, **71**) were isolated from a benzene extract of the “Meao” variant of cannabis from Thailand [27]. Three cannabichromene derivatives, (±)-4-acetoxycannabichromene (**72**), (±)-3”-hydroxy-Δ^4”^-cannabichromene (**73**) and (–)-7-hydroxycannabichromane (**74**), were isolated from high potency *C. sativa* using silica-gel VLC, normal-phase silica HPLC, and reverse-phase silica (C_18_) HPLC. These derivatives (**72**–**74**) were chemically identified using 1D and 2D NMR spectroscopic techniques [31]. In 1984, the CBC-C_3_ derivative (**75**) was reported, and using mass spectral analysis, the chemical structure of the CBC-C_3_ derivative (**75**) was determined to be 2-methyl-2-(4-methyl-2-pentyl)-7-propyl-2H-1-benzopyran-5-ol [53]. The structures of these nine CBC-type cannabinoids (**67**–**75**) are shown in Figure 5.

### 2.9. Cannabinol (CBN) Type (Eleven Compounds)

The chemical structure and the isolation details of seven cannabinol derivatives (**76**–**82**, Figure 5) have been reported in 1980 [3]. Both 8-hydroxycannabinol (**83**) and 8-hydroxy cannabinolic acid A (**84**) were isolated from the high potency variety of *C. sativa* and chemically identified based on NMR and high-resolution mass (HR-MS) analysis in 2009 [31]. The compounds 1′*S*-hydroxy-cannabinol (**85**) and 4-terpenyl cannabinolate (**86**) were isolated from the same cannabis variety (high potency *C. sativa*), and their chemical structures were confirmed by GC-MS analysis [18].

### 2.10. Cannabitriol (CBT) Type (Nine Compounds)

Nine CBT-type cannabinoids, including (−)-*trans*-CBT-C_5_ (**87**), (+)-*trans*-CBT-C_5_ (**88**), (±)-*cis*-CBT-C_5_ (**89**), (±)-*trans*-CBT-C_3_ (**90**), CBT-C_3_-homologue (**91**), (−)-*trans*-CBT-OEt-C_5_ (**92**), (–)-*trans*-CBT-OEt-C_3_ (**93**), 8,9-Di-OH-CBT-C_5_ (**94**), and CBDA-C_5_ 9-OH-CBT-C_5_ ester (**95**) (Figure 6), have been isolated from cannabis. Cannabitriol (**87**) was originally reported in 1966 from Japanese hemp [28], but its chemical structure was elucidated in 1976 [54]. The configuration of this compound was later determined by X-ray analysis [55]. The compounds (+)-*trans*-CBT-C_5_ (**88**) and (–)-*trans* -CBT-OEt-C_5_ (**92**) were isolated by ElSohly et al. from the ethanolic extract of cannabis, which was chromatographed on a silica gel column [56] and identified by GCMS.

The two ethoxy cannabitriols (**92**–**93)** are most likely artifacts, since ethanol was used in their isolation from *Cannabis* [3,57].

### 2.11. Miscellaneous Types Cannabinoids (30 Compounds)

A total of thirty miscellaneous type cannabinoids (Figure 7 and Figure 8) have been isolated from cannabis, including dehydrocannabifuran (DCBF-C_5_, **96**), cannabifuran (CBF-C_5_, **97**), 8-hydroxy-isohexahydrocannabivirin (OH-iso-HHCV-C_3_, **98**), 10-oxo-Δ^6a(10a)^-tetrahydro-cannabinol (OTHC, **99**), cannabicitran (**100**), (–)-Δ^9^-*cis*-(6a*S*,10a*R*)-tetrahydro-cannabinol (*cis*-Δ^9^-THC, **101**), cannabicoumaronone (CBCON-C_5_, **102**), cannabiripsol (CBR, **103**), cannabitetrol (CBTT, **104**), cannabichromanone-C_5_ (CBCN-C_5_, **105**), cannabichromanone-C_3_ (CBCN-C_3_, **106**),(±)-Δ^7^-*cis*-isotetrahydrocannabivarin-C_3_ (*cis*-iso-Δ^7^-THCV,**107**), (–)-Δ^7^-*trans*-(1*R*,3*R*,6*R*)-isotetrahydrocannabi- varin-C_3_ (*trans*-iso-Δ^7^-THCV,**108**), and (–)-Δ^7^-*trans*-(1*R*,3*R*,6*R*)-isotetrahydrocannabinol-C_5_ (*trans*-iso-Δ^7^-THC, **109**). The cyclohexane-methanol extract of Afghan hashish afforded dehydrocannabifuran (DCBF-C_5_, **96**), cannabifuran (CBF-C_5_, **97**), OTHC (**99**), and cannabichromanone-C_5_ (CBCN-C_5_, **105**) after micropreparative GC and TLC. Their chemical structures were determined by mass and NMR spectroscopic analysis [58]. Cannabicitran (CBT, **104**) was isolated from an ethanolic extract of Lebanese hashish. It was purified by counter-current distribution and silica gel chromatography. Its chemical structure was determined by GCMS, IR and ^1^H NMR analyses [59]. The isolation of *cis*-Δ^9^-THC (**101)** was reported from a petroleum extract of marihuana by Smith and Kampfert in 1977. The extract was purified on a florsil column followed by preparative TLC [60].

Cannabiripsol (CBR, **103**) was isolated from a South African Cannabis variant after hexane extraction and chromatography on silica and polyamide columns. Its chemical structure was determined by spectral means (IR, GCMS, UV, ^1^H NMR) and by synthesis [61]. Cannabitetrol (CBTT, **104**) was obtained and identified from the hexane extract of Mexican marijuana grown in Mississippi using silica gel column chromatography [62]. ElSohly and Slade reported the details of the isolation and chemical identification of compounds **106**–**109** [2].

From a high potency variety of *C. sativa**,* three cannabichromanones were isolated, namely cannabichromanone B (**110**), C (**111**), and D (**112**). The absolute configurations of these three cannabinoids were assigned on the basis of the Mosher ester and inspection of their circular dichroism spectra. The isolation of these compounds was performed using semi-preparative C_18_ HPLC [63]. Additionally, (–)-(7*R*)-cannabicoumarononic acid (**113**), 4-acetoxy-2-geranyl-5-hydroxy-3-*n*-pentylphenol (**114**), and 2-geranyl-5-hydroxy-3-*n*-pentyl-1,4-benzoquinone (**115**) were isolated from the buds and leaves of the same variety of cannabis (high potency *C. sativa*) using several chromatographic techniques, including silica-gel VLC, solid-phase extraction, reverse-phase columns (C_18_ SPE), and normal-phase HPLC. In addition, 5-acetoxy-6-geranyl-3-*n*-pentyl-1,4-benzoquinone (**116)** was isolated on silica gel column chromatography followed by normal-phase HPLC [32]. In 2010, a new cannabinoid named cannabimovone (CBM, **117**) was isolated from a non-psychotropic variety of *C. sativa*. This unusual metabolite is presumably formed from CBD and was isolated from a polar fraction of hemp by using flash chromatography over reverse-phase C_18_ silica gel followed by normal-phase HPLC. The chemical identity of CBM (**117**) was revealed by a combination of 1D and 2D NMR along with ESI-MS spectroscopic techniques [64]. A tetracyclic cannabinoid, cannabioxepane, (CBX, **118**) was isolated in 2011 from a cannabis variety called Carmagnola by applying many chromatographic techniques including RP-18 column, silica gel column chromatography, and NP-HPLC chromatography. Its chemical structure was established using MS and NMR data [65]. Seven more cannabinoids (**119**–**125**) were isolated from a high potency variety of *C. sativa* (Figure 7 and Figure 8) and chemically elucidated by 1D and 2D NMR and HRMS analyses as 10*α*-hydroxy-Δ^9,11^-hexahydrocannabinol (**119**), 9*β*,10*β*-epoxyhexahydrocannabinol (**120**), 9*α*-hydroxyhexahydrocannabinol (**121**), 7-oxo-9*α*-hydroxyhexa-hydrocannabinol, (**122**) 10*α*-hydroxyhexahydrocannabinol (**123**), 10a*R*-hydroxyhexahydrocannabinol (**124**), and 9*α*-hydroxy-10-oxo-Δ^6a,10a^-THC (**125**) [18,20].

## 3. Non-Cannabinoids

In addition to cannabinoids, more than 400 non-cannabinoid constituents have been isolated and/or identified from the cannabis plant. These non-cannabinoids belong to various chemical classes [2,3,66]. The major non-cannabinoid constituents are classified into four major categories: non-cannabinoid phenols, flavonoids, terpenes, alkaloids.

### 3.1. Non-Cannabinoid Phenols (42 Compounds)

Non-cannabinoid phenols include many chemical classes, including spiro-indans, dihydrostilbenes, dihydrophenathrenes, and simple phenols.

#### 3.1.1. Spiro-Indans (16 Compounds)

Sixteen spiro-indan type compounds were isolated from cannabis (**126**–**141**, Figure 9). Compound **126** was isolated in 1976 from an Indian cannabis variety using silica gel column chromatography and given the name cannabispiran [67]. In the same year, the compound was also isolated by another research group from a South African variety and named cannabispirone; cannabispirenone (**127**) was also identified from the South African cannabis variety [68]. Bercht et al. used NMR and mass spectrometric analysis to prove the chemical structures of **126** and **127**, while Ottersen et al. used X-ray crystallography to confirm the chemical structure of **126** [67]. The cannabispirenone isomer (**128**), with interchangeable methoxy and hydroxyl groups, was isolated from Mexican marihuana, and its chemical structure was established by ^1^H NMR and EIMS analysis, [69]. Cannabispiradienone (**129**) was isolated from Thai cannabis, and its chemical structure was elucidated based on ^1^H NMR spectroscopy and confirmed by hydrogenation to give cannabispiran (**126**) [70]. Two spiro-indans named cannabispirol (**130**) and acetyl cannabispirol (**131**) were detected by Yukihiro and Nishioka in the benzene extract of the dried leaves of Japanese cannabis. The benzene extract was chromatographed on a polyamide column followed by silica gel chromatography to yield compounds **130** and **131** [71]. Compound **130** was also isolated from a South African variety of cannabis grown in Mississippi by Turner’s group and named *β*-cannabispirol. The orientation of the hydroxyl group was determined by ^1^H NMR analysis of **130** and its acetate derivative (**131**) [61]. Three spiro-indans were isolated from an ethanolic extract of a seized hashish sample from Saudi Arabia and were chemically identified as 5-hydroxy-7-methoxyindan-1-spiro-cyclohexane (**132**), 7-hydroxy-5-methoxyindan-1-spiro-cyclohexane (**133**), and 5,7-dihydroxyindan-1-spiro-cyclohexane (**134**) through a combination of spectral and chemical analysis. The methanolic fraction of hashish was subjected to flash chromatography and further purified through silica gel column chromatography to afford compounds **132**–**134** [72]. Isocannabispiran (**135**) has been isolated from a Panamanian variety of cannabis by repeated chromatography. The structure was chemically elucidated as 5′-hydroxy-7′-methoxy-spiro-(cyclohexane-1,1′-indan)-4-one by spectroscopic means as well as direct comparison with cannabispiran (**126**) [73]. Radwan et al. (2008) isolated 7-O-methyl-cannabispirone (**136**) from an extract of a high potency cannabis variety using normal phase chromatography followed by C_18_-HPLC [32], while isocannabispiradienone (**137**) and *α*-cannabispiranol (**138**) were obtained from the dichloromethane extract of decarboxylated *C. sativa* hemp that was subjected to C_18_ flash chromatography, followed by silica gel gravity column chromatography and HPLC. The chemical structures were established using HR-ESIMS and NMR (^1^H, ^13^C, HSQC and HMBC) data [66]. Recently, three new cannabispirans (**139**–**141**) have been obtained from the leaves of *C. sativa*. Cannabispirketal (**139**) and the glycoside, *α*-cannabispiranol-4′-O-β-glucopyranose (**140**) were isolated from an ethanolic extract, and their chemical structures were determined by 1D NMR (^1^H NMR, ^13^C NMR) and 2DNMR (COSY, HSQC, HMBC and ROESY) [74]. In 2018, Nalli et al. isolated prenylspirodienone (**141**) and proved its chemical structure by extensive NMR and ESI-MS analysis. Moreover, they proposed the bio-synthetic pathway from this compound [75].

#### 3.1.2. Dihydrostilbenes (12 Compounds)

Twelve dihydrostilbenes (**142**–**153**) were isolated and identified from *C. sativa* (Figure 10). Turner et al. reported four new dihydrostilbenes in his review [3]. These compounds are 3-[2-(4-hydroxyphenyl)-ethyl]-5-methoxyphenol (**142**), 3-[2-(3-hydroxy-4-methoxyphenyl)-ethyl]-5-methoxyphenol (**143**), 3-[2-(3-isoprenyl-4-hydroxy-5-methoxy-phenyl)-ethyl]-5-methoxyphenol (**144**) and canniprene (**145**). Their chemical structures were determined by chemical and spectral analysis, while the chemical structure of **145** was confirmed by synthesis [71]. Two dihydrostilbenes, named cannabistilbene I (**146**) and cannabistibene II (**147**), were isolated from the polar acidic fraction of a Panamanian variant of *C. sativa* grown at the University of Mississippi. The chemical structure of cannabistilbene I (**146**) was elucidated as 3,4′-dihydroxy-5-methoxy-3′(3-methylbut-2-enyl)-dihydrostilbene by ^1^H NMR and mass spectral analysis and confirmed by total synthesis. The chemical structure of cannabistilbene II (**147**) was proposed as either **147A** or **147B** based on spectral analysis [76]. El-Feraly isolated 3,4′,5-trihydroxy-dihydrostilbene (**148**) from the ethanol extract of a hashish sample. The chemical structure of **148** and its prepared triacetate were established by ^1^H and ^13^C NMR spectroscopic analysis as well as total synthesis [77]. Recently, Guo et al. isolated three new prenylated stilbenes (**149**–**151**) and two other known stilbenes (**152**–**153**) from the leaves of *C. sativa* grown in Yunnan Province, China. The authors applied multiple chromatographic techniques in the isolation and purification of compounds **149**–**153**, such as column chromatography over silica gel cc, ODS C_18_ Si gel column chromatography, Sephadex column chromatography, and preparative HPLC. Spectral techniques such as 1D and 2D NMR spectroscopy and HRESIMS were used to determine the chemical structures as α,α′-dihydro-3′,4,5′-trihydroxy-4′-methoxy-3-isopentenylstilbene (149), α,α′-dihydro-3,4′,5-trihydroxy-4-methoxy-2,6-diisopentenylstilbene (150), and α,α′-dihydro-3′,4,5′-trihydroxy-4′-methoxy-2′,3-diisopentenylstilbene (151), α,α′-dihydro-3,4′,5-trihydroxy-4,5′-diisopentenylstilbene (152) and combretastatin B-2 (153) [78].

#### 3.1.3. Dihydrophenanthrenes (Seven Compounds)

Two dihydrophenenathrens, cannabidihydrophenanthrene (cannithrene 1) (**154**) and cannithrene 2 (**155**), were isolated from Thailand cannabis [71,79]. The chemical structure of cannithrene 2 was confirmed by X-ray crystallography of its diacetate derivative. The authors proposed that cannabidihydrophenanthrene (**154**) was biogenetically derived from cannabispiradienone (**129**) via a diene-phenol rearrangement [71]. Two dihydrophenanthrenes (**156**–**157**) and one phenethrene derivative (**158**) were isolated from an ethanolic extract of a high potency cannabis variety grown in Mississippi using a combination of normal and reversed phase chromatographic techniques and were identified as 4,5-dihydroxy-2,3,7-trimethoxy-9,10-dihydrophenanthrene (**156**), 4-hydroxy-2,3,6,7-tetramethoxy-9,10-dihydrophenanthrene (**157**) and 4,7-dimethoxy-1,2,5-trihydroxyphenanthrene (**158**) [80]. In 2008, Sanchez-Duffhues isolated the known 1,4-phenanthrenequinone, denbinobin (**159**) from an acetone extract of *C. sativa* chemotype (CARMA) after fractionation and column chromatography. Denbinobin (**159**) was purified by crystallization from ether [81]. Another 1,4-phenanthrenequinone derivative was also isolated from the leaves and branches of *C. sativa*, its chemical structure was determined as 2,3,5,6-tetramethoxy 9,10-dihydrophenanthrenedione (**160**) using 1D and 2D NMR as well as ESI-MS [82]. The chemical structures of the reported dihydrophenathrenes and phenanthrenes are presented in Figure 11.

#### 3.1.4. Simple Phenols (Seven Compounds)

Five simple phenols (**161**–**167**) were detected in the essential oil of Cannabis (Figure 12) and identified by GC/MS as eugenol (**161**), methyleugenol (**162**), iso-eugenol (**163**), trans-anethol (**164**) and *cis*-anethol (**165**) [3,83]. Vanillin (**166**) was isolated and identified from hemp pectin using silica gel column chromatography and identified via ^1^H NMR, ^13^C NMR, and ESI-MS spectroscopic methods [84]. Phloroglucinol β-D-glucoside (**167**) was identified from the stem exudate of greenhouse-grown *C. sativa* by TLC, but its aglycone (phloroglucinol) was isolated after acid hydrolysis of the exudate. Phloroglucinol was identified by ^1^H NMR and GC/MS experiments [85].

### 3.2. Flavonoids (34 Compounds)

Thirty-four flavonoids were isolated from *C. sativa* (**168**–**201**), which could be categorized into seven basic chemical skeletons that can be methylated, glycosylated (C or O glycosides), prenylated, or geranylated (Figure 13). The seven chemical structures of the flavonoid aglycones are orientin, vitexin, isovitexin, apigenin, luteolin, kaempferol and quercetin. The details of the isolation and chemical structures of 19 flavonoids (**168**–**171**, **173**–**175**, **178**–**184**, **187**–**188**, and **193**–**195**) isolated from *C. sativa* were reviewed by turner et al. in 1980 [3]. The flavonoid glycosides, vitexin (**172**), cytisoside (**176**) and cytisoside glucoside (**177**), were identified from Canadian cannabis plants grown from seeds, where the authors used TLC, a hydrolytic test and UV spectroscopic analysis to determine their chemical structures [86]. Crombie et al. isolated two methylated isoprenoid flavones named Canniflavone 1 and Canniflavone 2 from a Thailand strain of cannabis [79]; four years later, Barrett et al. isolated the same two compounds and named them Canniflavin A (**189**, prenyl flavone) and Canniflavin B (**190**, geranyl flavone) from the ethanolic extract of *C. sativa*. The structures were elucidated by using UV, ^1^H NMR and ^13^C NMR spectroscopic techniques [87]. In 2008, Radwan el al isolated another methylated isoprenoid flavone, canniflavin C (**191**), along with 6-prenylapigenin (**185**) and chrysoeriol (**192**), from a high potency variety of *C. sativa* grown in Mississippi polar fractions by using combination of various chromatographic techniques, such as VLC, silica gel column chromatography, and RP-HPLC. The geranyl moiety in Canniflavin C (**191**) is attached to the C-8 instead of the C-6 as in Canniflavin B based on 1D and 2D NMR analysis [80]. The glycoside apigenin-6,8-di-C-β-D-glucopyranoside (**186**) was isolated from the methanolic extract of hemp [88], Two flavonoid glycosides (**196**–**197**) were isolated from the pollen grains of the male plants of a Mexican variety of *C. sativa* that was cultivated at the University of Mississippi. Their chemical structures were identified as kaempferol-3-O-sophoroside (**196**) and quercetin-3-O-sophoroside (**197**) based on 1D and 2D NMR and UV experiments [89]. The flavonoid glycoside Rutin (**198**) was isolated for the first time from hemp pectin. The ethanolic extract was purified by macroreticular resin, silica gel column chromatography, and Sephadex-LH-20. Spectroscopic methods (ESI-MS, ^1^H NMR, ^13^C NMR) were used for identification of its chemical structure [84].

The flavonoids Quercetin (**199**), Naringenin (**200**), and Naringin (**201**) were identified and quantified in the hydroalcoholic extract of hemp inflorescence from monoecious cultivars grown in Central Italy. Four cultivars (Ferimon, Uso-31, Felina 32 and Fedora) were analyzed at four stages of growth from flowering to ripening using HPLC-PDA. Naringenin (**200**) and its glycoside (Naringin, **201**) was detected only in Fedora 17 and Ferimon cultivars, respectively, while Quercetin (**199**) was present in the hydroalcohlic extract of the four cultivars [90,91]. Naringenin (**200**) was also detected and quantified in the industrial hemp of Futura 75 cultivar cultivated in Italy and the quantification was performed by HPLC-DAD-MS of the water extract [91].

### 3.3. Terpenes (120 Compounds)

Previous publications have reported 120 [2,92] or more [93,94] terpenes in cannabis. Our search for terpenes with the correct chemical structures yielded a total of 120 terpenes. Terpenes, or isoprenoids, consist of the second largest class of cannabis constituents. These compounds are responsible for the characteristic aroma of the plant. Terpenes can be classified into five main classes: monoterpenes, sesquiterpenes, diterpenes, triterpenes, and miscellaneous terpenes. Out of 120 terpenes, there are 61 monoterpenes (C_10_ skeleton), 51 sesquiterpenes (C_15_ skeleton), 2 diterpenes, (C_20_ skeleton), 2 triterpenes (C_30_ skeleton), and 4 miscellaneous compounds.

#### 3.3.1. Monoterpenes (61 Compounds)

Sixty-one monoterpenes (**202**–**262**) have been reported from cannabis. They can be classified into two distinct classes: monoterpene hydrocarbons and oxygenated monoterpenes. Cannabis monoterpenes may be acyclic (linear), monocyclic or bicyclic (Figure 14 and Figure 15).

In 1942, the low boiling point terpene fraction of Egyptian hashish contained two monoterpene hydrocarbons: *p*-cymene (**205**) and small quantities of 1-methyl-4-isopropenyl-benzene or dehydro-*p*-cymene (**208**) [95]. Almost twenty years later, an acyclic monoterpene, myrcene (**202**) and the monocyclic monoterpene limonene (**209**), were identified in the essential oil of fresh, wild *C. sativa* from Canada [96].

Terpenoid-related compounds were detected from the hydrodistillation of freshly harvested *C. sativa* L. from India [97]. The essential oil obtained from the hydrodistillation underwent fractional distillation, yielding five fractions. Fraction 5 was further chromatographed over alumina using petroleum ether, benzene, ether, and alcohol successively as eluents. The fractions collected with petroleum ether were combined and named Fraction 5-A, while the fractions collected with benzene as the solvent system were combined and collectively known as Fraction 5-B. Upon GC-MS and physico–chemical analyses, twenty-four different metabolites were detected, out of which 12 monoterpenes were not reported previously, namely α-terpinene (**207**), β-phellandrene (**208**), γ-terpinene (**209**), α-terpinolene (**210**), α-pinene (**214**), β-pinene (**215**), camphene (**216**), linalool (**221**), α-terpineol (**232**), terpinene-4-ol (**233**), linalool oxide (**243**), and sabinene hydrate (**249**) [97].

In the early 1970s, Dutch and Turkish cannabis volatile oil samples were compared by capillary gas chromatography [98,99]. The volatile oils were prepared by two methods: hydrodistillation or through nitrogen extraction; the volatile oils were identified to contain 18 total monoterpenes, seven of which had not been identified previously: *cis*-β-ocimene (**203**), *trans*-β-ocimene (**204**), α-phellandrene (**211**), Δ^3^-carene (**217**), Δ^4^-carene (**218**), sabinene (**219**) and α-thujene (**220**). In 1976, the oxygenated monoterpene, *m*-mentha-1,8-(9)-dien-5-ol (**229**) was identified from the volatile oil of *Cannabis* [100].

As the aroma of *C. sativa* L. is of high importance for the detection of illicit marijuana trafficking, the composition of the emitted aroma constituents was investigated, using a direct gas chromatographic analysis of the headspace components. The marijuana standard was grown and harvested at the University of Mississippi, and real samples were obtained from Customs’ seizures. The samples were prepared by weighing 1 g of each sample, placing it in a microvial, and heating the sample at 65 °C for 1 h. From the microvial, 5 mL of the headspace air was withdrawn using a gas-tight syringe and directly injected into the gas chromatograph. A total of 18 terpenic compounds were detected, out of which three compounds had not been previously reported, namely 2-methyl-2-heptene-6-one (**227),** fenchyl alcohol (**254**), and borneol (**256**), [101].

The volatile oil of *C. sativa* of Mexican origin was prepared, and a total of 17 monoterpenes were identified in the oil through GC-MS analysis. Six oxygenated monoterpenes had not been identified previously, namely nerol (**223**), geraniol (**224**), carvacrol (**230**), 1,8-cineol (**250**), 1,4-cineol (**251**), and camphor (**258**) [60]. Later that year, piperitenone (**231**) was detected and identified in the volatile oil of *Cannabis* through GC-MS analysis and retention time matching [102].

A total of 55 monoterpenes were identified by the Bos research group in both 1975 and 1978, with 24 being reported for the first time from cannabis including the monoterpene hydrocarbon, 3-phenyl-2-methyl-prop-1-ene (**213**), and 23 oxygenated hydrocarbons, namely citral B (**222**), citronellol (**226**), geranyl acetone (**228**), carvone (**231**), pulegone (**235**), dihydrocarvone (**236**), β-terpineol (**237**), dihydrocarveyl acetate (**238**), *p*-cymene-8-ol (**239**), β-cyclocitral (**241**), safranal (**242**), *cis*-linalool oxide (**244**), perillene (**245**), sabinol (**246**), thujyl alcohol (**247**), piperitone oxide (**252**), piperitenone oxide (**253**), fenchone (**255**), bornyl acetate (**257**), camphene hydrate (**259**), α-pinene oxide (**260**), pinocarveol (**261**), and pinocarvone (**262**), [83,103]. The chemical structure of the identified monoterpenes were shown in Figure 14 and Figure 15. 

Ross and ElSohly studied the composition of cannabis essential oil prepared by steam distillation of marijuana buds, by collecting the oil using the lighter-than-water volatile oil apparatus. The study included the preparation of volatile oil from fresh buds, as well as samples that were dried and stored at room temperature at three time points: 1 week, 1 month, and 3 months. The composition of the oils was then determined by GC-MS and GC-FID, where the percentage compositions of each component determined by the GC-FID method, and the presence of the terpenes was identified by GC-MS. There was a total of 68 components identified in the study, with 57 identified as monterpenes, sesquiterpenes, and other compounds, such as ketones and esters. The fresh buds’ oil was determined to have the greatest composition of monoterpenes, at approximately 92.48%. Sesquiterpenes represented 6.84% of the total percent composition and other compounds represented only 0.68%. Analysis of the three-month samples showed a decrease in the percent composition of monoterpenes and an increase in the percent composition of sesquiterpenes and other compounds, where the monoterpenes represented 62.02% composition, the sesquiterpenes were 35.63%, and other compounds represented 2.35% of the volatile oil. Three new oxygenated monoterpenes were reported for the first time in this study, namely, ipsdienol (**225**), *cis*-carveol (**240**) and *cis*-sabinene hydrate (**248**) [104].

#### 3.3.2. Sesquiterpenes (51 Compounds)

In 1942, a study was conducted for the analysis of the higher boiling point fraction of Egyptian hashish, resulting in the identification of one sesquiterpene, α-caryophyllene (α-humulene) (**263**) [95]. This was the first sesquiterpene to be identified in cannabis. Around 20 years later, in addition to α-caryophyllene (**263**), β-caryophyllene (**264**) was identified in the volatile oil of fresh *C. sativa*, through GC analysis [105]. Five sesquiterpenes were identified in the volatile oil of Indian *C. sativa* in 1965. These sesquiterpenes were, namely, caryophyllene oxide (**265**), curcumene (**266**), α-*trans*-bergamotene (**267**), α-selinene (**268**), and β-farnesene (**269**), [97]. In 1973, four new sesquiterpenes were reported from the analysis of the volatile oil of *C. sativa*, namely longifolene (**270**), humulene epoxide I (**271**), humulene epoxide II (**272**), and caryophyllene alcohol (caryophyllenol) (**273**) [106]. The sesquiterpene β-bisabolene (**274**) has only been reported in one study, which included the analysis of headspace volatiles, volatile oil, and samples of marijuana from Customs’ seizures [101]. Another three sesquiterpenes, allo-aromadendrene (**275**), calamenene (**276**), and α-copaene (**277**), were reported for the first time from the essential oil of *C. sativa* grown in Mexico in 1974. The compounds were identified using both GC/FID and GC/MS [102].

The volatile oil of *C. sativa* from Mexico was studied by Bercht and Paris in 1974, and a total of 17 monoterpenes were identified in the oil through GC-MS analysis. A sesquiterpene, nerolidol (**278**), was identified for the first time [60]. Using GC/MS and GC retention time, α-gurjunene (**279**) was detected for the first time in *C. sativa* resin [102].

In 1975, four sesquiterpenes were identified in the essential oil of *Cannabis*, namely iso-caryophyllene (**280**), β-selinene (**281**), selina-3,7(11)-diene (**282**), and selina-4(14),7(11)-diene (**283**) [103]. This sesquiterpene was confirmed by the same research group to be present in the essential oil of *C. sativa* in 1978 by GC and GC-MS analyses [83]. In addition to α-gurjunene (**279**), other sesquiterpenes were also reported for the first time in this study were: α-bisabolol (**284**), α-cedrene (**285**), α-cubebene (**286**), δ-cadinene (**287**), epi-β-santalene (**288**), farnesol (**289**), γ-cadinene (**290**), γ-elemene (**291**), γ-eudesmol (**292**), guaiol (**293**), ledol (**294**), *trans*-*trans*-α-farnesene (**295**), (*Z*)-β-farnesene (**296**) and farnesyl acetone (**297**) [83]. In 1996, 14 new sesquiterpenes were identified, namely α-cadinene (**298**), α-*cis*-bergamotene (**299**), α-eudesmol (**300**), α-guaiene (**301**), α-longipinene (**302**), α-ylangene (**303**), β-elemene (**304**), β-eudesmol (**305**), epi-α-bisabolol (**306**), γ-*cis*-bisabolene (**307**), γ-curcumene (**308**), γ-muurolene (**309**), γ-*trans*-bisabolene (**310**), and viridiflorene (**311**) [104]. Recently, germacrene-B (**312**) was detected for the first time from hemp essential oil and was quantified by GC-MS, while clovandiol (**313**) was identified in organic extract of cannabis infloresence of Ferimon and Uso-31 cultivars [90,107]. The chemical structures of the reported sesequiterpenes are shown in Figure 16.

#### 3.3.3. Diterpenes

Phytol (**314**) and neophytadiene (**315**) are the only two triterpenes reported from *C. sativa* (Figure 17). They were identified by GC-MS [83,90].

#### 3.3.4. Triterpenes

Two triterpenes have been identified in cannabis (Figure 17). In 1971, analysis of the ethanolic extract of *Cannabis* roots resulted in the identification of two triterpenes, friedelin (friedelan-3-one, **316**) and epifriedelanol (**317**), via spectral data and comparison with authentic samples [108].

#### 3.3.5. Miscellaneous Terpenes

A total of four miscellaneous terpenes have been identified in cannabis (Figure 17). Two of them, vomifoliol (**318**) and dihydrovomifoliol (**319**), are isophorone-type compounds and were isolated from Dutch hemp [60]. Both were identified from the stems and leaves of the plant through isolation, spectral data comparison, and synthesis from (+)-α-ionone. The other two miscellaneous-type compounds were identified from the volatile oil of *C. sativa*, namely β-ionone (**320**) and dihydroactinidiolide (**321**) [83].

### 3.4. Alkaloids

Only two spermidine alkaloids (**322**–**323**) have been identified in *C. sativa* (Figure 18). In 1975, the first spermidine alkaloid was isolated from a methanolic extract of cannabis roots from a Mexican variant grown in Mississippi, and it was identified as cannabisativine (**316**) by X-ray crystallography [109]. Later the same year, the same compound was isolated from an ethanolic extract of the dry leaves and small stems of a Thailand variant [110]. The ethanolic extract was extracted and acid–base partitioned, as well as subjected to column and thin-layer chromatography followed by crystallization of the alkaloid from acetone.

A year after cannabisativine was reported, the second spermidine alkaloid, namely anhydrocannabisativine (**323**), was isolated from the dry leaves and small stems of cannabis of the Mexican variety grown in Mississippi [111]. The compound was isolated through a series of acid–base extractions and silica-gel chromatography. The identity of the compound was proven by spectral data analysis and by the conversion of cannabisativine (**322**) to anhydrocannabisativine (**323**). In 1978, the spermidine alkaloid was isolated from the roots and leaves of a Mexican variant [112]. In 1977, the Mississippi group also identified anhydrocannabisativine (**322**) in 15 different *Cannabis* variants using TLC eluted with chloroform: acetone: ammonia (1:1:1) [113].

## 4. Conclusions

To date, there have been over 500 constituents of *Cannabis sativa* reported, out of which 125 are identified to belong to the cannabinoid-type compounds, with five new cannabinoids reported in the last 2 years. The other non-cannabinoid-type compounds are classified into various chemical classes, including alkaloids, flavonoids, non-cannabinoid phenols, and terpenes. This review discusses the chemistry, identification, isolation, and structural elucidation of these major classes of compounds, to provide an overview of their chemical structures and to better understand the complexity of the chemical profile of *C. sativa*.

## Figures and Tables

**Figure 1 molecules-26-02774-f001:**
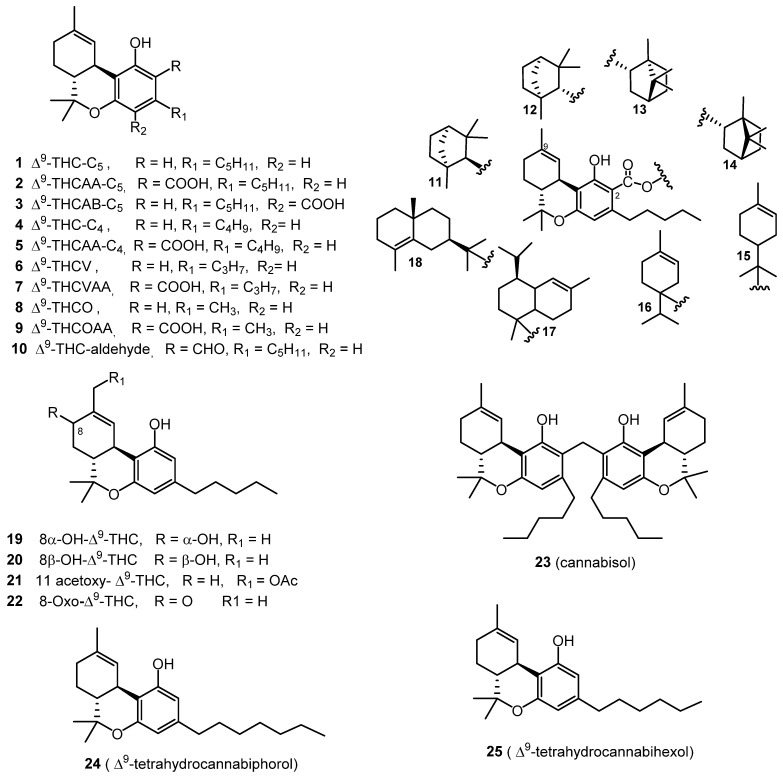
Δ^9^-THC-Type Cannabinoids.

**Figure 2 molecules-26-02774-f002:**
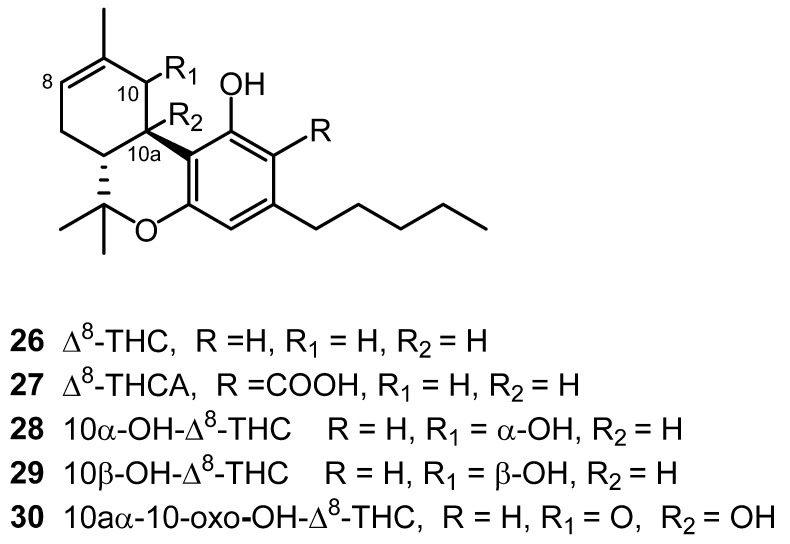
Δ^8^-THC-Type Cannabinoids.

**Figure 3 molecules-26-02774-f003:**
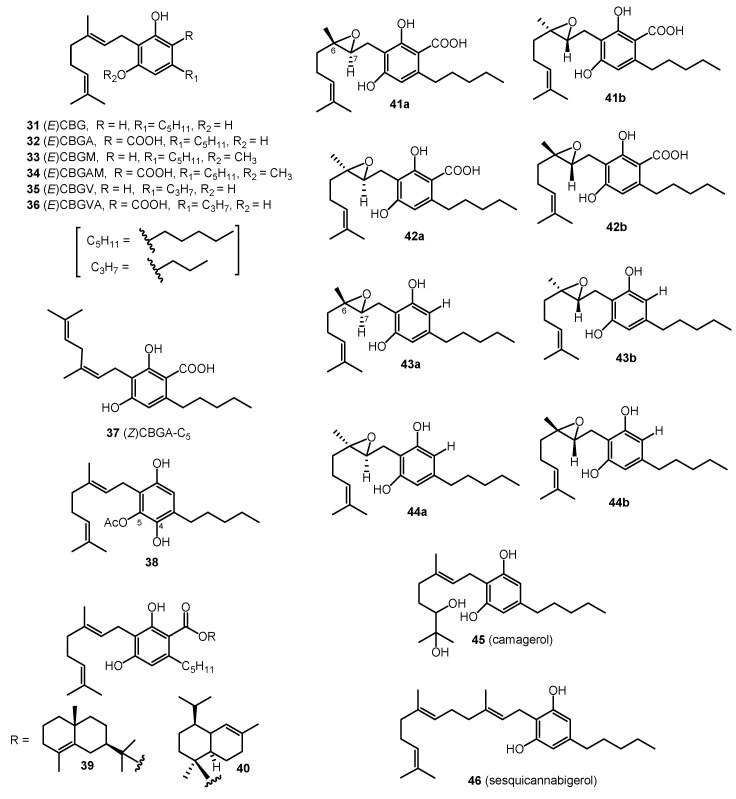
CBG-Type Cannabinoids.

**Figure 4 molecules-26-02774-f004:**
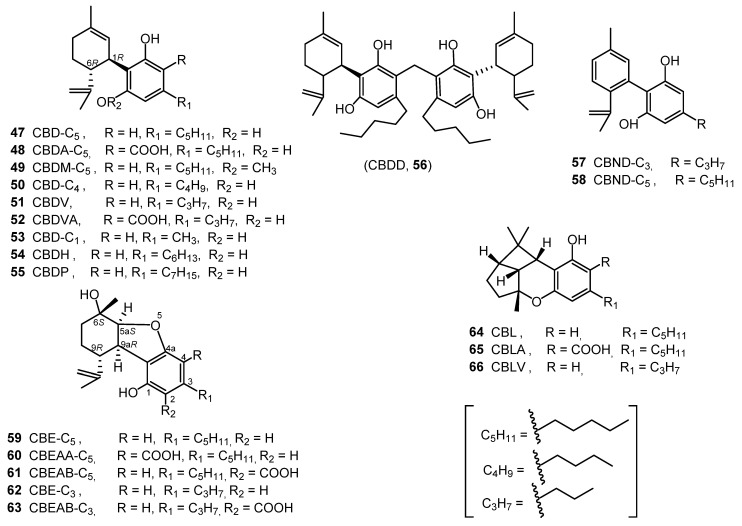
CBD, CBND, CBE and CBL-Type Cannabinoids.

**Figure 5 molecules-26-02774-f005:**
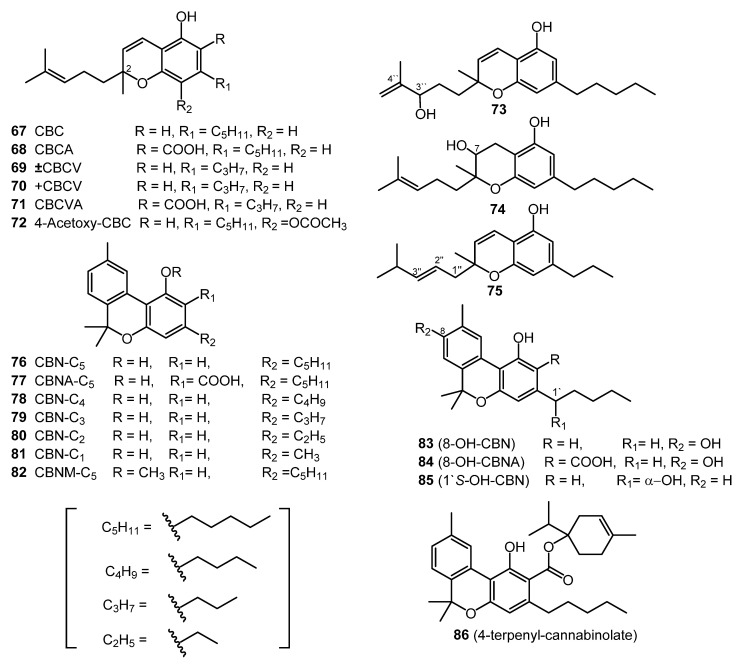
CBC and CBN-Type Cannabinoids.

**Figure 6 molecules-26-02774-f006:**
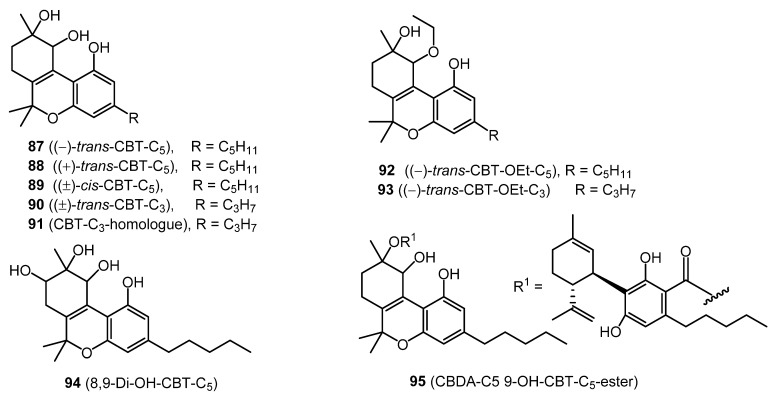
CBT-Type Cannabinoids.

**Figure 7 molecules-26-02774-f007:**
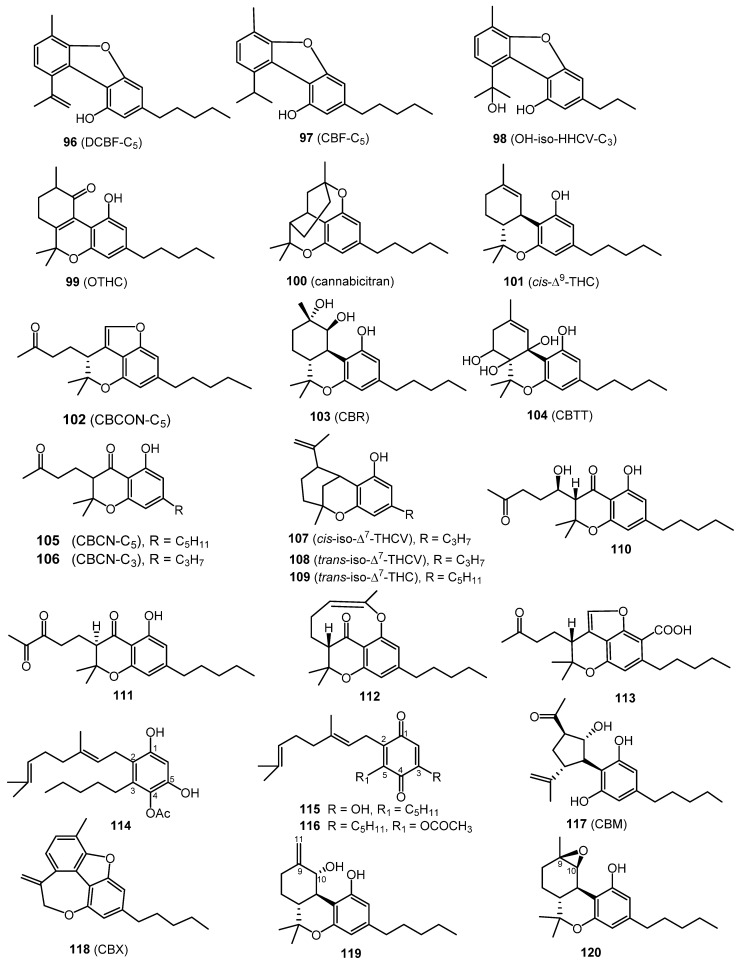
Miscellaneous-Type Cannabinoids.

**Figure 8 molecules-26-02774-f008:**
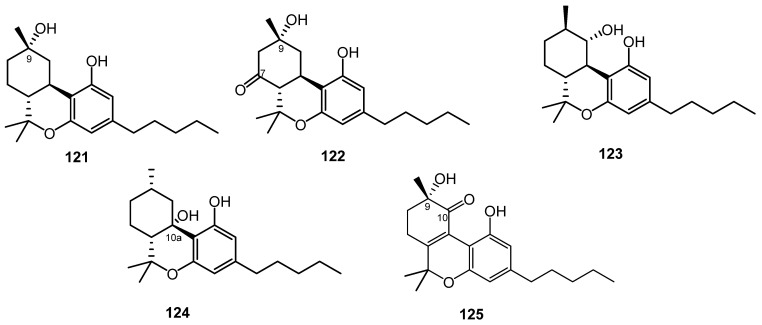
Miscellaneous-Type Cannabinoids (cont.).

**Figure 9 molecules-26-02774-f009:**
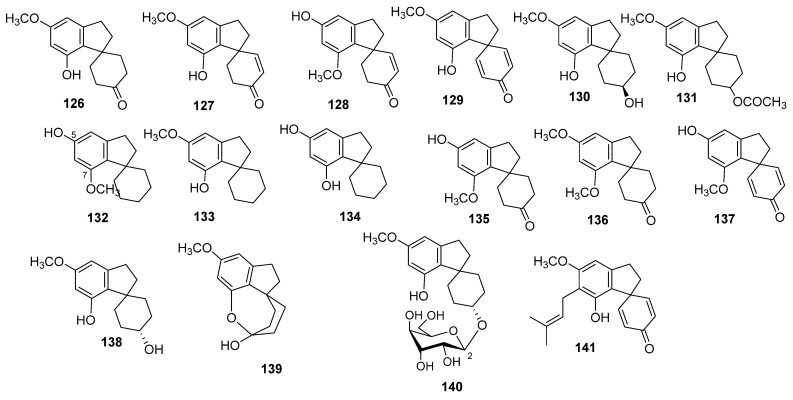
Chemical structure of Cannabis Spiro-Indans.

**Figure 10 molecules-26-02774-f010:**
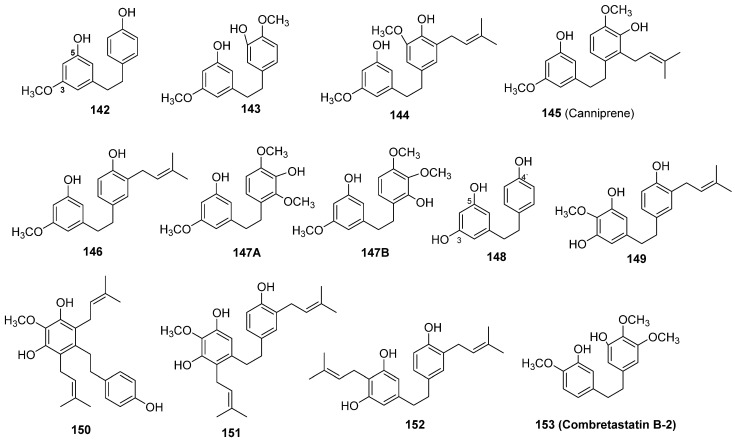
Chemical structure of Cannabis dihydrostilbenes.

**Figure 11 molecules-26-02774-f011:**
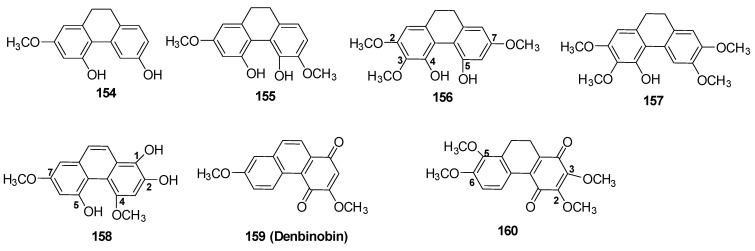
Chemical structure of Cannabis Dihydrophenanthrene Derivatives.

**Figure 12 molecules-26-02774-f012:**
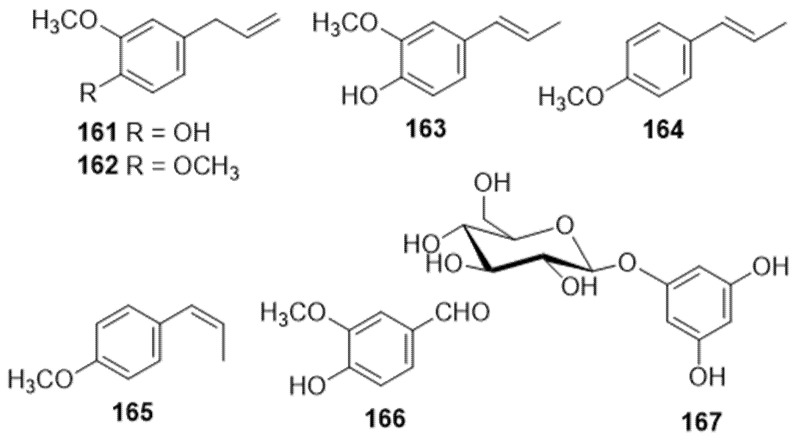
Chemical Structures of Cannabis Simple Phenols.

**Figure 13 molecules-26-02774-f013:**
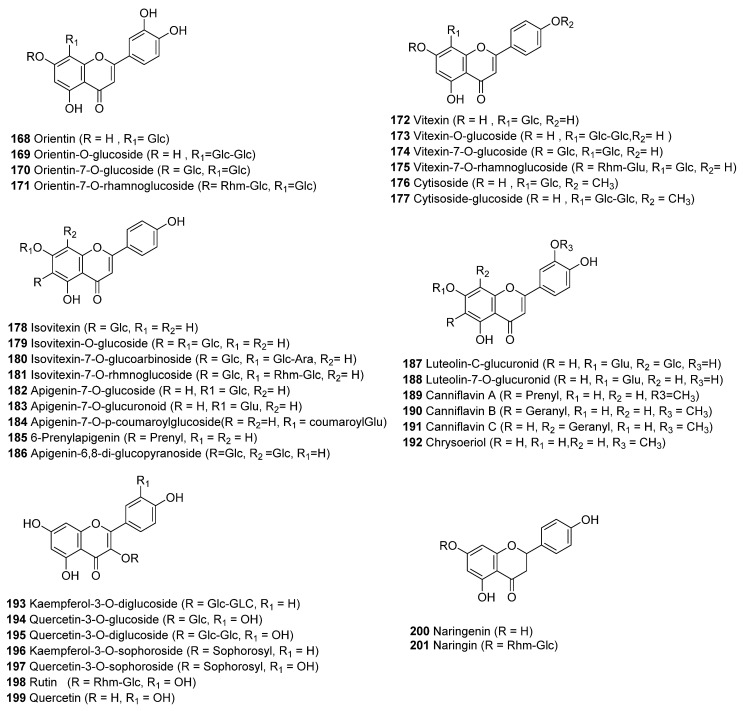
Chemical Structures of Cannabis Flavonoids.

**Figure 14 molecules-26-02774-f014:**
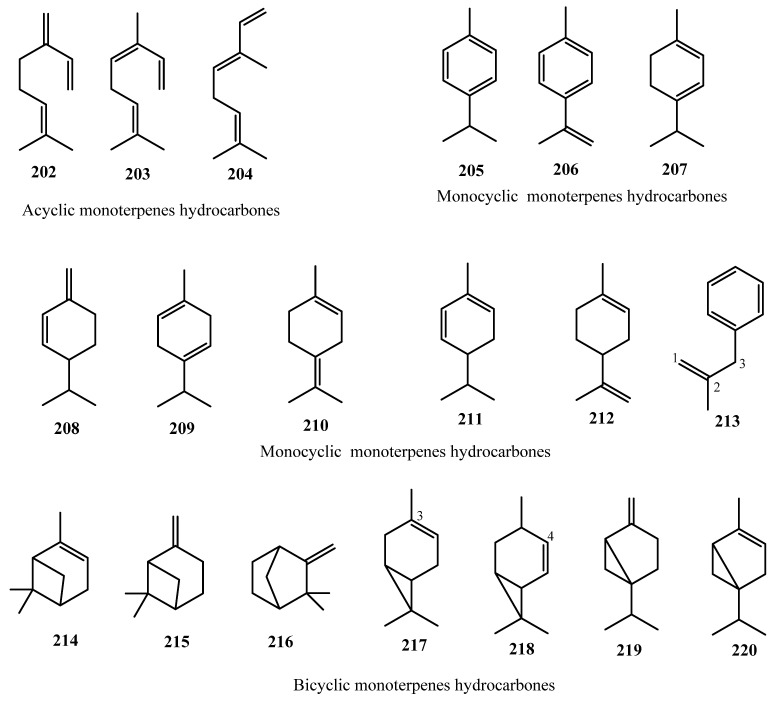
Chemical Structure of Cannabis Monoterpenes (hydrocarbon derivatives).

**Figure 15 molecules-26-02774-f015:**
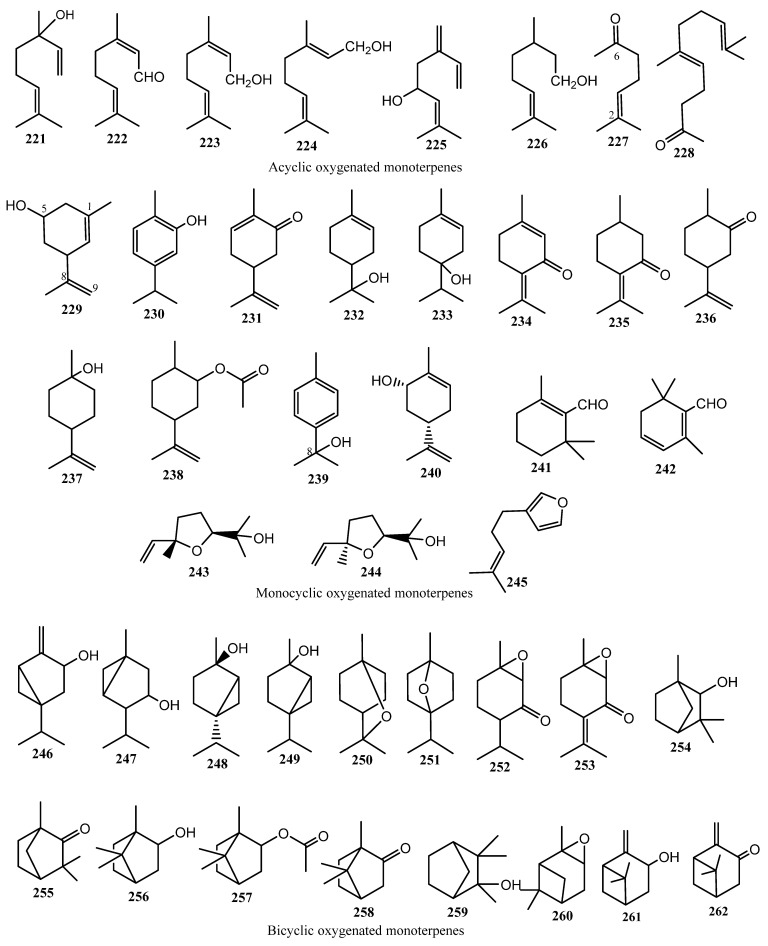
Chemical Structure of Cannabis Oxygenated Monoterpenes (acyclic, monocyclic and bicyclic).

**Figure 16 molecules-26-02774-f016:**
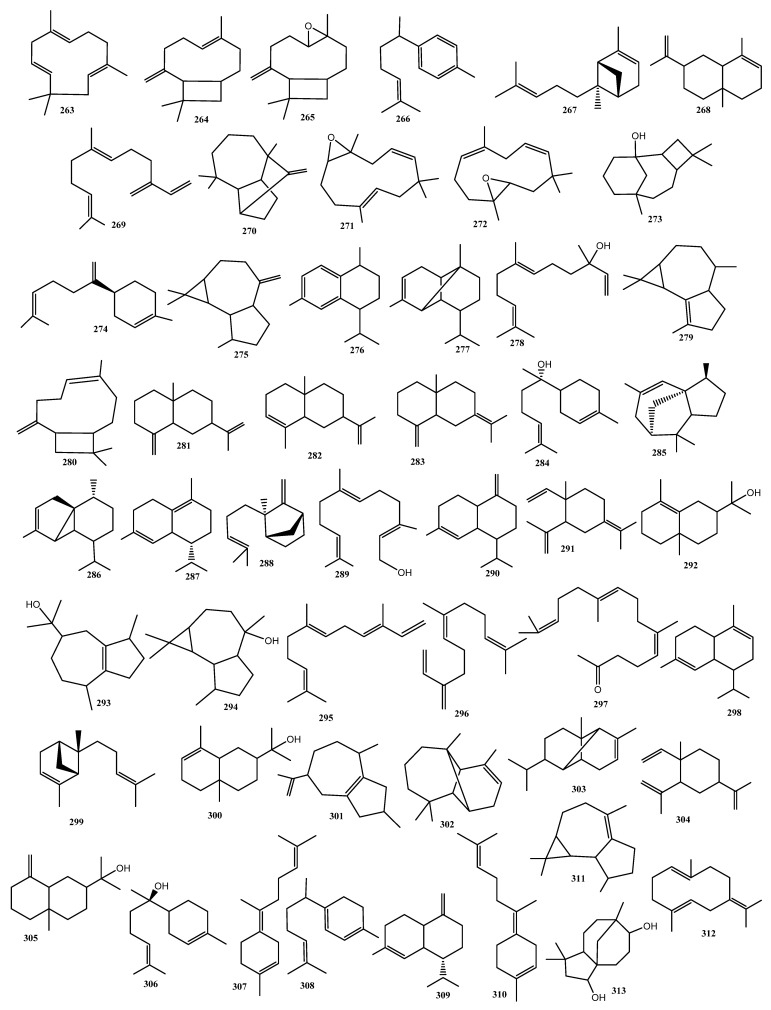
Chemical Structures of Cannabis Sesquiterpenes.

**Figure 17 molecules-26-02774-f017:**
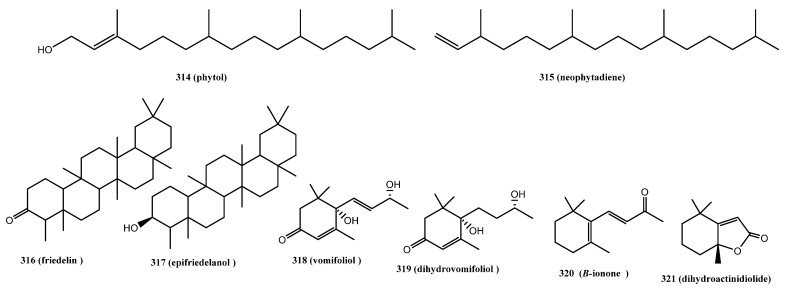
Chemical Structures of Cannabis Diterpenes, Triterpenes, and Miscellaneous Terpenes.

**Figure 18 molecules-26-02774-f018:**
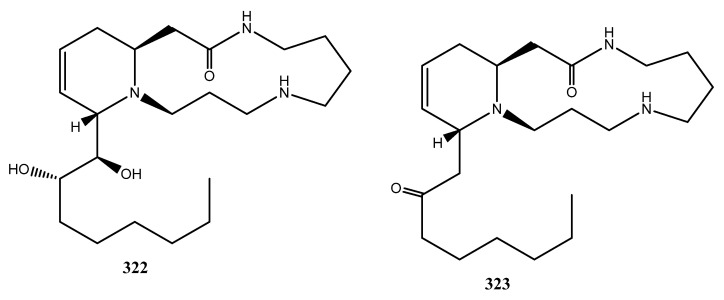
Chemical structure of Cannabis Alkaloids.

## Data Availability

Not applicable.

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
