# Peer review of "Cannabinoids, Phenolics, Terpenes and Alkaloids of Cannabis"

_molecules, 2021, doi:10.3390/molecules26092774_

Round 1
Reviewer 1 Report
A nice review dealing with the large plethora of phytometabolites discovered in an emerging plant used for different purposes. I think that this kind of approach to describe the content of a natural matrix is suitable for the readers. The language can be improved.
Just few suggestions before approval:
line 15: mssing dot
lines 23, 24 and 45: please italicize the species and genus names
line 30: change to "occur"
line 39: change to "required"
lines 60, 61, 63, 68, 117: correct "tetrahydrcannabinol"
text in general: hexane, hexanes, n-hexane. I suggest to make it uniform
line 81: ether or diethyl ether?
line 211: change to "Its structure"
line 214: correct "hepyl"
line 215: I prefer "hexane" or "hexanic" extract
line 218: correct "fcartions"
line 219: change to "CBDJ and CBDP were"
line 220: correct "sterioselective"
line 222: correct "si gel"
line 239: correct "Labenese"
line 258: cannabinoma?
line 264: change to "The carboxylic acid derivative of CBC"
line 335: correct "4-actoxy"
line 347: the name of the variety is "Carmagnola"
line 385: change to "South African"
line 413: correct "Dihydrostibenes"
line 427: correct "dihydrosilbene"
Figure 13: correct "glucouronoid" (twice)
line 532: change to "physico-chemical"
line 580: change to "other compounds"
Did the authors check also the following references to add new compounds found in this plant?
Menghini L, Ferrante C, Carradori S, D'Antonio M, Orlando G, Cairone F, Cesa S, Filippi A, Fraschetti C, Zengin G, Ak G, Tacchini M, Iqbal K. Chemical and Bioinformatics Analyses of the Anti-Leishmanial and Anti-Oxidant Activities of Hemp Essential Oil. Biomolecules. 2021 Feb 12;11(2):272. doi: 10.3390/biom11020272.
di Giacomo V, Recinella L, Chiavaroli A, Orlando G, Cataldi A, Rapino M, Di Valerio V, Politi M, Antolini MD, Acquaviva A, Bacchin F, Di Mascio M, Leone S, Brunetti L, Menghini L, Carradori S, Zengin G, Ak G, Ferrante C. Metabolomic Profile and Antioxidant/Anti-Inflammatory Effects of Industrial Hemp Water Extract in Fibroblasts, Keratinocytes and Isolated Mouse Skin Specimens. Antioxidants (Basel). 2021 Jan 1;10(1):44. doi: 10.3390/antiox10010044.
Ingallina C, Sobolev AP, Circi S, Spano M, Fraschetti C, Filippi A, Di Sotto A, Di Giacomo S, Mazzoccanti G, Gasparrini F, Quaglio D, Campiglia E, Carradori S, Locatelli M, Vinci G, Rapa M, Ciano S, Giusti AM, Botta B, Ghirga F, Capitani D, Mannina L. Cannabis sativa L. Inflorescences from Monoecious Cultivars Grown in Central Italy: An Untargeted Chemical Characterization from Early Flowering to Ripening. Molecules. 2020 Apr 20;25(8):1908. doi: 10.3390/molecules25081908.
Zengin G, Menghini L, Di Sotto A, Mancinelli R, Sisto F, Carradori S, Cesa S, Fraschetti C, Filippi A, Angiolella L, Locatelli M, Mannina L, Ingallina C, Puca V, D'Antonio M, Grande R. Chromatographic Analyses, In Vitro Biological Activities, and Cytotoxicity of Cannabis sativa L. Essential Oil: A Multidisciplinary Study. Molecules. 2018 Dec 10;23(12):3266. doi: 10.3390/molecules23123266.
Orlando G, Recinella L, Chiavaroli A, Brunetti L, Leone S, Carradori S, Di Simone S, Ciferri MC, Zengin G, Ak G, Abdullah HH, Cordisco E, Sortino M, Svetaz L, Politi M, Angelini P, Covino S, Venanzoni R, Cesa S, Menghini L, Ferrante C. Water Extract from Inflorescences of Industrial Hemp Futura 75 Variety as a Source of Anti-Inflammatory, Anti-Proliferative and Antimycotic Agents: Results from In Silico, In Vitro and Ex Vivo Studies. Antioxidants (Basel). 2020 May 17;9(5):437. doi: 10.3390/antiox9050437.
Reviewer 2 Report
Cannabinoids, Phenolics, Terpenes and Alkaloids of Cannabis
In this review the authors discussed the complex chemistry of C. sativa, focusing on identification, isolation, and structural elucidation of major classes of constituents. The article is well written and comprehensive. However I have a major observation. When discussing the isolated components the authors never refer to the quantification issue. In my opinion it is equally important to identify the components, but also to know the detected amount. Moreover, the article is quite long. This is very good from one point of view, especially for the readers interested in relevant details, but some others are (generally) in a hurry. To solve both issues, I suggest the authors to create a big table, where they should insert all the detected components in the first column, and in the next ones they can summarize the extraction techniques, methods used or analyses, detected amounts & other relevant information, and finally in the last column they can cite the references.
Line 45 – Please write C. sativa in italics
Please mention at the end of the introduction the aim of the article and also its originality. Motivate, by comparing with the other review articles on the same topic, why it is still a room for a new review about C. sativa.
Round 2
Reviewer 2 Report
The authors operated the changes and/or justified well their responses. I suggest the acceptance of the manuscript in the present form.